# Neoadjuvant Therapy in Robotic Lung Surgery: Elevating Surgical Complexity Without Compromising Outcomes

**DOI:** 10.3390/cancers16233938

**Published:** 2024-11-25

**Authors:** Inés Serratosa, Carlos Déniz, Camilo Moreno, Iván Macia, Francisco Rivas, Anna Muñoz, Marina Paradela, Ernest Nadal, Miguel Mosteiro, Susana Padrones, Marta García, Tania Rodríguez-Martos, Judith Marcè, Amaia Ojanguren

**Affiliations:** 1Department of Thoracic Surgery, Hospital Universitari de Bellvitge, L’Hospitalet de Llobregat, 08907 Barcelona, Spain; iserratosa@bellvitgehospital.cat (I.S.);; 2Bellvitge Institute for Biomedical Research, L’Hospitalet de Llobregat, 08907 Barcelona, Spain; 3Department of Medical Oncology, Catalan Institute of Oncology, L’Hospitalet de Llobregat, 08907 Barcelona, Spain; 4Department of Respiratory Medicine, Hospital Universitari de Bellvitge, L’Hospitalet de Llobregat, 08907 Barcelona, Spain

**Keywords:** lung cancer, neoadjuvant therapy, robotic assisted thoracic surgery

## Abstract

Neoadjuvant therapy, including chemotherapy and chemo-immunotherapy, is becoming an established strategy to enhance oncological outcomes for patients with locally advanced non-small cell lung cancer (NSCLC). This study assessed whether receiving neoadjuvant therapy affects the results of robotic-assisted thoracic surgery, a minimally invasive approach that may reduce pain and recovery time. By comparing patients who received neoadjuvant therapy with those who did not, we found that although neoadjuvant treatment slightly extended operative time, it did not increase rates of major postoperative complications or conversion to open surgery. Our findings suggest that robotic surgery remains a safe and feasible option for NSCLC patients undergoing neoadjuvant therapy, supporting its use as an effective surgical technique that may facilitate recovery.

## 1. Introduction

In recent years, neoadjuvant therapies, particularly chemoimmunotherapy, have transformed the treatment paradigm for locally advanced and resectable non-small cell lung cancer (NSCLC). The introduction of immune checkpoint inhibitors (ICIs), such as nivolumab and pembrolizumab, into the neoadjuvant setting has revolutionized treatment, offering higher rates of major pathological response (MPR) and complete pathological response (pCR) compared to chemotherapy alone [1,2,3,4]. Clinical trials like CheckMate 816 [5] and NADIM II [6] have demonstrated that preoperative chemoimmunotherapy significantly improved survival outcomes and reduced residual viable tumors at the time of surgery. Notably, these advancements are reshaping the treatment landscape, with neoadjuvant therapy now being considered the standard of care for patients with resectable stage II and III NSCLC.

Neoadjuvant therapy, however, presents surgical challenges such as tumor fibrosis and hilar adhesions, which may heighten the likelihood of conversion to open surgery [7,8,9]. Bott et al. reported that more than half of the minimally invasive procedures were converted to thoracotomy due to fibrosis and inflammation induced by neoadjuvant nivolumab [10]. Additionally, neoadjuvant-treated patients may reach surgery in a more vulnerable state, with increased operative complexity potentially leading to extended anesthesia exposure and a higher risk of complications. These findings emphasize the critical need for the careful selection of surgical techniques and thorough preoperative assessment to address these complexities effectively.

Robotic-assisted thoracic surgery (RATS) offers advantages in precision, dexterity and visualization, enabling complex procedures with minimal invasion. This results in reduced postoperative pain, shorter hospital stays, and faster recovery compared to both open surgery and video-assisted thoracoscopic surgery [11,12]. RATS may also help mitigate the surgical challenges posed by neoadjuvant therapy, such as fibrosis and adhesions, improving overall outcomes [13,14]. Evidence suggests RATS is effective and safe for patients treated with chemoimmunotherapy, and may offer superior outcomes compared to VATS, including reduced Intensive Care Unit stays and more accurate staging [15,16]. 

As neoadjuvant therapies increasingly render previously considered unresectable tumors operable, understanding their effects on surgical outcomes in robotic-assisted procedures becomes essential. This study aims to assess the impact of neoadjuvant therapy on surgical outcomes by using propensity score matching to compare patients who received neoadjuvant treatment with those who did not.

## 2. Material and Methods

This retrospective cohort study includes patients who underwent robotic-assisted thoracic surgery (RATS) for non-small cell lung cancer (NSCLC) at a tertiary referral cancer center between February 2019 and August 2024. 

### 2.1. Inclusion Criteria

Patients were included in this study based on the following criteria: histologically confirmed non-small cell lung cancer (NSCLC), resectable disease (clinical stages I to III), and planned anatomical pulmonary resections (segmentectomy, lobectomy, or pneumonectomy), with complete clinical data available. Patients with non-primary lung tumors, unresectable tumors (stage IV), or those undergoing minor resections were excluded from the analysis. 

A total of 391 patients met these inclusion criteria, 23 of whom received neoadjuvant therapy prior to robotic-assisted thoracic surgery (RATS). Neoadjuvant regimens included platinum-based chemotherapy alone or in combination with immunotherapy (Nivolumab or Pembrolizumab), as determined by multidisciplinary tumor board recommendations. Among patients receiving neoadjuvant therapy, the median time from the final neoadjuvant treatment to surgery was 23 ± 4 days.

### 2.2. Endpoints 

The primary endpoint of this study was to evaluate the association between neoadjuvant therapy and the incidence of postoperative complications. The postoperative complications analyzed include prolonged air leak (>5 days), pneumothorax, subcutaneous emphysema, pleuro-pulmonary infection, surgical wound infection, respiratory failure, hemothorax, myocardial infarction, postoperative atrial fibrillation, ischemia or stroke, phrenic nerve injury, bronchopleural fistula, chylothorax, reoperation, readmission within 30 days, and mortality.

The secondary endpoints of this study were first, a comparison of operative time and conversion rates to open surgery between patients who received neoadjuvant therapy and those who did not; second, an analysis of differences in hospital stay duration and chest drainage time between the two treatment groups; and third, an evaluation of whether the type of neoadjuvant therapy (chemotherapy alone versus chemoimmunotherapy) influenced the incidence of postoperative complications.

### 2.3. Propensity Score Matching

A propensity score matching analysis was conducted to compare surgical outcomes between patients who received neoadjuvant therapy and those who did not. The goal was to minimize baseline differences, improving comparability and reducing selection bias. A 1:10 matching was performed based on key preoperative variables such as age, gender, body mass index (BMI), pulmonary function tests, and comorbidities. Propensity scores were calculated using a logistic regression model, and matched pairs were identified using the nearest-neighbor algorithm without replacement, with a caliper width of 0.2. The final matched cohort included 253 patients (23 neoadjuvant, 230 non-neoadjuvant). The balance between groups was verified by comparing means, proportions, and standardized mean differences (SMD), with an SMD below 0.1 indicating good balance [17].

### 2.4. Statistical Analysis

Descriptive statistics were used to summarize baseline characteristics, with continuous variables expressed as means ± standard deviation and categorical variables as frequencies and percentages. Intraoperative outcomes, such as surgical time and conversion rates to open surgery, were compared between the matched groups using weighted linear regression and the McNemar test, respectively. Weights were applied due to group size imbalance (1:10 matching). Postoperative complications were analyzed using the McNemar test to evaluate differences between groups, with odds ratios (OR) and 95% confidence intervals calculated to assess the association between neoadjuvant therapy and complication incidence. The McNemar test was chosen for its suitability in paired categorical data analysis, allowing the detection of marginal differences between matched groups. Given the number of complications assessed, we acknowledge the risk of Type I errors but focused on interpreting findings with the highest clinical relevance and statistical significance. The hospital stay and chest drainage durations were compared using the Wilcoxon signed-rank test. Fisher’s exact test was used to compare complications between chemotherapy and chemoimmunotherapy groups, while surgical time was assessed using the *t*-Student test. A *p*-value < 0.05 was considered statistically significant, and all analyses were conducted using IBM-SPSS Statistics v.20.

### 2.5. Ethical Statement

The study adhered to the Declaration of Helsinki and was approved by the Ethics Committee of Hospital Universitari de Bellvitge, given its retrospective design using anonymized patient data.

## 3. Results

### 3.1. Baseline Characteristics and Propensity Score Matching

The final matched cohort included 253 patients with a mean age of 67.24 years, of whom 73.52% were male. Pulmonary function showed a mean forced expiratory volume in one second (FEV1) of 86.12% and a diffusing capacity of the lungs for carbon monoxide (DLCO) of 80.07%. Comorbidities were common, with 30.83% having hypertension, 13.83% diabetes, and 26.88% dyslipidemia. Smoking history varied, with 17.39% being active smokers and 36.36% former smokers. These preoperative characteristics are summarized in detail in Table 1, providing a comprehensive overview of the matched cohort’s clinical profile. 

Table 2 presents the results of the Propensity Score Matching (PSM) analysis comparing neoadjuvant and non-neoadjuvant groups. Variables matched included demographic characteristics, pulmonary function, and comorbidities. The standardized mean difference (SMD) was used to assess balance, with values below 0.1 indicating good balance. Age, sex, lung function, and most comorbidities were well-balanced between groups, though BMI (SMD = 0.23) and atrial fibrillation (SMD = 0.19) showed moderate imbalances. Overall, despite some differences, the baseline characteristics were sufficiently balanced to allow for valid comparisons of surgical outcomes, supporting the study’s conclusions.

### 3.2. Operative Outcomes

A weighted linear regression was used to compare surgical time between the neoadjuvant and control groups, adjusting for the 1:10 matching ratio. The neoadjuvant group had a mean surgical time of 250.0 ± 66.63 min, 28.57 min longer than the control group (221.4 ± 71.03 min), with a statistically significant coefficient (*p* = 0.001). The R-squared value of 0.043 indicates that neoadjuvant therapy explains 4.3% of the variance in surgical time, showing an association between neoadjuvant treatment and increased surgical time.

The McNemar test compared conversion rates to open surgery, which were 8.7% in the neoadjuvant group and 3.9% in the control group. With a *p*-value of 0.5, the results indicated no statistically significant difference between the two groups.

### 3.3. Postoperative Complications

The McNemar test was used to assess postoperative complications between the neoadjuvant and non-neoadjuvant groups. Significant differences were found for several complications, as shown in Table 3. The neoadjuvant group had a higher rate of prolonged air leaks (39.13% vs. 35.21%, *p* < 0.001) and postoperative atrial fibrillation (8.7% vs. 2.54%, *p* = 0.03). In contrast, the non-neoadjuvant group had higher rates of pneumothorax, respiratory insufficiency, subcutaneous emphysema, pleural or pulmonary infections, hemothorax, and ischemia or stroke, all showing significant differences (*p* < 0.05). Wound infections showed no significant difference. 

A conditional logistic regression was conducted to assess the impact of neoadjuvant therapy on specific complications, including prolonged air leak, subcutaneous emphysema, pleuro-pulmonary infection, and postoperative atrial fibrillation. None of these complications showed statistically significant differences between the neoadjuvant and non-neoadjuvant groups, as seen in Table 4. The odds ratios for air leak (OR = 1.22), subcutaneous emphysema (OR = 0.60), and pleuro-pulmonary infection (OR = 0.71) were not significant, with wide confidence intervals including 1. Although postoperative atrial fibrillation showed a potential increased risk in the neoadjuvant group (OR = 4.26, *p* = 0.07), it did not reach statistical significance.

### 3.4. Hospital Stay and Drainage Duration

The mean hospital stay was 5.96 days (SD = 3.30) for the neoadjuvant group and 6.08 days (SD = 3.96) for the non-neoadjuvant group. A Wilcoxon signed-rank test (W = 110.5, *p* = 0.86) showed no statistically significant difference in hospital stay between the two groups.

The mean drainage duration was 5.96 days (SD = 4.06) for the neoadjuvant group and 5.71 (SD = 4.28) days for the non-neoadjuvant group. A Wilcoxon signed-rank test (W = 128.0, *p* = 0.76) showed no statistically significant difference, indicating that neoadjuvant therapy does not significantly impact drainage time.

### 3.5. Chemotherapy Alone vs. Chemoimmunotherapy

Of the 23 patients who received neoadjuvant therapy, 12 were treated with chemoimmunotherapy and 11 with chemotherapy alone.

The mean surgical time was similar between the chemotherapy group (249.73 min) and the chemo-immunotherapy group (250.25 min), with a *t*-test (*p* = 0.98) indicating no significant difference based on the type of preoperative treatment. Conversion rates to thoracotomy were also comparable between the two groups (9.09% vs. 8.33%, *p* = 1.0). Table 5 reflects Fisher’s exact test results, which showed no significant differences in postoperative complications between the groups, though air leaks lasting more than five days were more frequent in the chemo-immunotherapy group (58% vs. 18%, *p* = 0.09) but not statistically significant.

Some complications could not be fully compared due to the absence of events in one or both groups.

## 4. Discussion

In recent years, the incorporation of neoadjuvant immunotherapy has transformed the treatment landscape for locally advanced resectable non-small cell lung cancer (NSCLC). Several randomized trials (CheckMate 816, NADIM II, AEGEAN, Keynote671 and Neotorch) have consistently demonstrated that neoadjuvant chemoimmunotherapy improves pathological response rates, increase resectability, and increase the cure rate and long-term survival outcomes [5,6,18,19,20]. However, the impact of these therapies on postoperative outcomes, particularly in minimally invasive procedures such as robotic-assisted thoracic surgery (RATS), remains less clearly defined. This study aimed to address that gap, employing propensity score matching to provide a robust comparison of patients treated with and without neoadjuvant therapy.

The results of this study, designed with the rigorous application of propensity score matching to minimize bias, suggest that neoadjuvant therapy, although associated with longer operative times and a higher incidence of prolonged air leaks, does not significantly impact conversion rates to open surgery or other key postoperative outcomes, such as hospital stay duration and chest drainage time. These findings are consistent with the evolving role of neoadjuvant therapy in thoracic surgery, as highlighted in recent studies like CheckMate77T [21], which demonstrated substantial improvements in pathological response rates without compromising surgical feasibility.

Our findings align with previous studies showing that, while neoadjuvant therapy is associated with longer operative times, it does not significantly increase conversion rates to thoracotomy or major postoperative complications [10,22,23]. The extended operative time observed in the neoadjuvant group likely reflects the added procedural complexity, as neoadjuvant treatment often results in greater tumor fibrosis and hilar adhesions, complicating the surgery. This prolonged operative time, though statistically significant, may also carry clinical implications; extended anesthesia exposure can heighten the risk of adverse events, especially in patients with comorbidities. Although we observed no increase in postoperative complications or conversion rates, these findings underscore the importance of optimizing operative duration to mitigate potential anesthesia-related risks. This observation is further supported by the CheckMate 816 trial, which found that neoadjuvant chemoimmunotherapy with nivolumab similarly led to longer operative times due to increased surgical complexity, but without a significant increase in conversion rates to open surgery [5]. These findings reflect the growing expertise in managing complex, minimally invasive surgeries after systemic treatments, highlighting the feasibility of performing RATS even in the context of neoadjuvant therapy.

The precision of robotic systems has been highlighted as a key factor in maintaining low conversion rates in complex cases. Studies like the one by Romero Román et al. reinforce this point, demonstrating that the advanced capabilities of robotic-assisted thoracic surgery (RATS) help reduce the need for conversion to thoracotomy, even in cases where neoadjuvant therapy has been administered [24]. Our results are consistent with these observations, as the conversion rates in our study did not differ significantly between the neoadjuvant (8.7%) and non-neoadjuvant (3.9%) groups, indicating that RATS can effectively manage the surgical challenges posed by neoadjuvant therapy.

Our study found no statistically significant differences in overall postoperative complications between the neoadjuvant and non-neoadjuvant groups, consistent with findings from previous studies by Zeng et al. and Gao et al., which also reported similar safety profiles in patients undergoing RATS following neoadjuvant therapy [15,16]. The higher incidence of prolonged air leaks in the neoadjuvant group (39.13% vs. 35.21%) is a significant finding and mirrors the observations from major trials such as NADIM II and AEGEAN, where neoadjuvant nivolumab plus chemotherapy was associated with similar increases in postoperative complications, including prolonged air leaks [6,18]. This complication, along with the observed rise in postoperative atrial fibrillation, underscores the potential need for targeted perioperative management in patients receiving neoadjuvant therapy, as prolonged air leaks can increase morbidity and atrial fibrillation poses additional cardiovascular risks, particularly in extensive resections. However, the air leaks observed in our study did not result in extended hospital stays or other significant complications, underscoring the importance of improved perioperative management for patients undergoing surgery following chemoimmunotherapy.

Contrary to expectations, there was no significant difference in hospital stay duration between the neoadjuvant and non-neoadjuvant groups (5.96 vs. 6.08 days, *p* = 0.86), aligning with Nelson et al., who reported similar durations regardless of neoadjuvant therapy [11]. This suggests that, despite the increased surgical complexity, postoperative recovery remains comparable, supporting the safety and feasibility of RATS. Although the *p*-values from the Wilcoxon signed-rank test did not indicate statistically significant differences in hospital stay, these findings may still hold clinical relevance. Even non-significant trends in recovery times can guide clinicians in anticipating patient needs and refining perioperative management to further optimize recovery outcomes. Incorporating ERAS protocols into thoracic surgery has shown to enhance recovery by reducing postoperative pain, improving early mobilization, and enhancing pulmonary function—all crucial factors in optimizing patient outcomes [25,26]. These benefits are particularly relevant for patients undergoing complex procedures, such as RATS, following neoadjuvant chemoimmunotherapy. Given the additional challenges these patients face, such as increased surgical complexity and immunotherapy-induced tissue changes, ERAS protocols provide a structured framework to mitigate these risks and facilitate faster recovery.

Additionally, chest drainage duration was also similar between groups (5.96 vs. 5.71 days, *p* = 0.76), consistent with findings from Huang et al., further highlighting the precision of robotic surgery in mitigating potential adverse effects from neoadjuvant therapy [13]. Together, these findings highlight the effectiveness of robotic surgery combined with ERAS protocols in managing patients undergoing lung resections, even in the context of complex neoadjuvant treatments.

Our comparison between chemotherapy alone and chemo-immunotherapy showed no significant differences in operative times or conversion rates. This aligns with other real-world studies, which also report comparable safety profiles for both neoadjuvant approaches, with no significant increase in perioperative complications [5,27]. Although some studies observed a higher rate of pathological response with chemo-immunotherapy, they also noted a slight increase in certain high-grade adverse events, underscoring the need for further research to refine patient selection and minimize risks [28,29,30]. The literature remains limited to direct comparisons between these two modalities in RATS, but studies like Duan et al. suggest that chemo-immunotherapy may slightly increase specific complications, such as atrial fibrillation, as observed in our study [31].

## 5. Limitations

This study’s retrospective design introduces potential limitations, particularly regarding selection biases and unmeasured confounding factors. Although propensity score matching was utilized to balance baseline characteristics and reduce bias, residual confounding cannot be entirely excluded. Additionally, the relatively small sample size of the neoadjuvant group limits the statistical power of our findings. As a single-center analysis, our results may also reflect center-specific practices and may not be fully generalizable. Further large-scale prospective studies are essential to validate these results and explore long-term outcomes for patients undergoing RATS following neoadjuvant therapy.

## 6. Conclusions

In conclusion, our study suggests that neoadjuvant therapy, despite increasing operative time, does not significantly impact conversion rates to open surgery or major postoperative complications in robotic-assisted lung surgery. Overall, RATS remains a safe and feasible approach for patients undergoing neoadjuvant treatment, but further large-scale studies are essential to confirm these findings and optimize neoadjuvant strategies, ultimately improving long-term survival for patients with resectable NSCLC.

## Figures and Tables

**Table 1 cancers-16-03938-t001:** Demographic and clinical data.

Variable		N	%	Mean	SD
Age (years)				67.24	9.36
BMI (Kg/m^2^)				27.34	3.69
FEV1 (%)				86.12	15.02
DLCO (%)				80.07	15.24
FVC (%)				99.13	13.79
Gender	Male	186	73.52		
	Female	67	26.48		
Hypertension		78	30.83		
Diabetes mellitus		35	13.83		
Dyslipidemia		68	26.88		
COPD		29	11.46		
Nephropathy		10	3.95		
Vasculopathy		22	8.70		
Ischemic heart disease		12	4.74		
Atrial fibrillation		4	1.58		
Smoking status	Non-smoker	13	5.14		
	Former smoker > 6 months	92	36.36		
	Current smoker	44	17.39		

BMI: Body Mass Index; FEV1: Forced Expiratory Volume in the first second; DLCO: Diffusing Capacity of the Lung for Carbon Monoxide; FVC: Forced Vital Capacity; COPD: Chronic Obstructive Pulmonary Disease.

**Table 2 cancers-16-03938-t002:** Baseline characteristics balance after PSM.

Variable		Neoadjuvant Group (n = 23)	Non-Neoadjuvant Group (n = 230)	SMD
Age (years)		64.22 ± 7.45	63.75 ± 9.49	0.04
BMI (Kg/m^2^)		28.57 ± 6.12	27.22 ± 3.35	0.23
FEV1 (%)		85.41 ± 15.59	86.19 ± 15.0	−0.13
DLCO (%)		79.98 ± 16.76	80.08 ± 15.12	0.01
FVC (%)		99.30 ± 14.13	99.11 ± 13.79	−0.06
Sex	Male	17 (73.9%)	82 (68.9%)	0.11
	Female	6 (26.1%)	37 (31.1%)	−0.11
Hypertension		13 (56.5%)	68 (57.1%)	−0.07
Diabetes mellitus		6 (26.1%)	27 (22.7%)	−0.05
Dyslipidemia		11 (47.8%)	52 (43.7%)	0.04
COPD		6 (26.1%)	25 (21.0%)	−0.05
Nephropathy		2 (8.7%)	13 (10.9%)	0.01
Vasculopathy		5 (21.7%)	20 (16.8%)	−0.1
Ischemic heart disease		3 (13%)	15 (12.6%)	0.03
Atrial fibrillation		2 (8.7%)	9 (7.6%)	0.19
Smoking status	non-smoker	1 (4.3%)	10 (8.4%)	−0.05
	Current smoker	15 (65.2%)	71 (59.7%)	−0.05
	Active smoker	7 (30.4%)	38 (31.9%)	−0.11

BMI: Body Mass Index; FEV1: Forced Expiratory Volume in the first second; DLCO: Diffusing Capacity of the Lung for Carbon Monoxide; FVC: Forced Vital Capacity; COPD: Chronic Obstructive Pulmonary Disease; SMD: Standardized Mean Difference.

**Table 3 cancers-16-03938-t003:** Postoperative complications.

Complication	Neoadjuvant Group (n = 23)	Non-Neoadjuvant Group (n = 230)	χ^2^	*p*
Air leak > 5 days	9 (39.13%)	125 (35.21%)	100.42	<0.001
Pneumothorax	0	28 (7.89%)	28	<0.001
Subcutaneous emphysema	1 (4.35%)	26 (7.32%)	23.15	<0.001
Pleural/pulmonary infection	1 (4.35%)	23 (6.48%)	20.17	<0.001
Wound infection	0	1 (0.28%)	1	0.32
Respiratory insufficiency	0	7 (1.97%)	7	0.01
Hemothorax	0	5 (1.41%)	5	0.03
Myocardial infarction	0	0		
Postoperative atrial fibrillation	2 (8.7%)	9 (2.54%)	4.45	0.03
Ischemia/Stroke	0	6 (1.69%)	6	0.01
Phrenic nerve injury	0	2 (0.56%)	2	0.16
Bronchio-pleural fistula	0	5 (1.41%)	5	0.03
Chylothorax	0	4 (1.13%)	4	0.05
Reintervention	0	5 (1.41%)	5	0.03
Readmission within 30 days	0	24 (6.76%)	24	<0.001
Mortality	0	4 (1.13%)	4	0.05

**Table 4 cancers-16-03938-t004:** Effect of neoadjuvant therapy on the incidence of complications.

Complication	OR (IC)	*p*
Air leak > 5 days	1.22 [0.52, 2.86]	0.65
Subcutaneous emphysema	0.6 [0.08, 4,62]	0.63
Pleural/pulmonary infection	0.71 [0.09, 5.44]	0.74
Postoperative atrial fibrillation	4.26 [0.87, 20.92]	0.07

**Table 5 cancers-16-03938-t005:** Postoperative complications in the neoadjuvant group.

Complication	Chemotherapy (n = 11)	Chemoimmunotherapy (n = 12)	*p*
Air leak > 5 days	2 (18%)	7 (58%)	0.09
Pneumothorax	0	0	1
Subcutaneous emphysema	0	1 (8%)	1
Pleural/pulmonary infection	0	1 (8%)	1
Wound infection	0	0	1
Respiratory insufficiency	0	0	1
Hemothorax	0	0	1
Myocardial infarction	0	0	1
Postoperative atrial fibrillation	1 (9%)	1 (8%)	1
Ischemia/Stroke	0	0	1
Phrenic nerve injury	0	0	1
Bronchio-pleural fistula	0	0	1
Chylothorax	0	0	1
Reintervention	0	0	1
Readmission within 30 days	0	0	1
Mortality	0	0	1

## Data Availability

The data are not publicly available due to privacy restrictions but can be requested from the corresponding author with the appropriate institutional approval.

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
