# Peer review of "Neoadjuvant Therapy in Robotic Lung Surgery: Elevating Surgical Complexity Without Compromising Outcomes"

_cancers, 2024, doi:10.3390/cancers16233938_

Round 1
Reviewer 1 Report
Comments and Suggestions for Authors
Serratosa et al reported their work named “Neoadjuvant Therapy in Robotic Thoracic Surgery: Elevating Surgical Complexity Without Compromising Outcomes” and concluded “The use of neoadjuvant therapy does not seem to have a substantial impact on postoperative out-32 comes following lung resection via RATS supporting its feasibility in selected patients. Neverthe-33 less, additional research is needed to fully validate the safety and feasibility of this approach in a 34 wider patient cohort.”. I have the following comments:
- Please edit title to have “lung” instead of “thoracic”.
- Please try to add a supplementary table with a list of published meta-analyses on neoadjuvant immunotherapy including number of included studies, patients, and outcomes.
- Please spell out any abbreviations at their first time, if possible eg NADIM
- Abstract/manuscript: Please move number of included patients from the methods section to the results in the abstract.
- Please specify the median time from receiving neoadjuvant immunotherapy vs chemotherapy till surgery.
- Please try to add love plot showing SMD before after matching.
- Please confirm that generative AI as ChatGPT or Copilot wasn’t used or at least was used and double-checked. This should be acknowledged if used,
- Please try to combine Table 1 and 2 in one table. Please try to compare groups before matching.
- Please report the statistical power of your primary outcome.
- Please describe the matching method eg nearest neighbor method without replacement and the used caliper.
Author Response
Comment 1: Please edit title to have “lung” instead of “thoracic”.
Response 1: Thank you for your suggestion. We have revised the title to replace "thoracic" with "lung" to reflect the specific focus on lung cancer in our study.
Comment 2: Please try to add a supplementary table with a list of published meta-analyses on neoadjuvant immunotherapy including number of included studies, patients, and outcomes.
Response 2: Thank you for your suggestion to include a supplementary table summarizing relevant meta-analyses. We have created the table as requested, which you will find attached. However, we believe this addition may not directly enhance the value of our study, as our primary focus is on assessing whether neoadjuvant therapy impacts surgical outcomes in robotic thoracic surgery, rather than on evaluating pathological response or survival outcomes post-treatment.
Comment 3: Please spell out any abbreviations at their first time, if possible eg NADIM
Response 3: Thank you for this observation. We have reviewed the manuscript and spelled out all abbreviations at their first mention to ensure clarity for the readers. However, for certain study names, such as "NADIM," we believe it is unnecessary to spell them out, as they are proper names referencing specific studies and do not hinder the readability of our manuscript.
Comment 4: Abstract/manuscript: Please move number of included patients from the methods section to the results in the abstract.
Response 4: Thank you for pointing this out. We have made the necessary changes (lines 35-37) to enhance the clarity of the abstract.
Comment 5: Please specify the median time from receiving neoadjuvant immunotherapy vs chemotherapy till surgery.
Response 5: Thank you very much for this insightful comment. We agree that specifying the median time from neoadjuvant treatment to surgery adds valuable detail and enhances the thoroughness of our study. We have now included this information in the Inclusion criteria section (page 3, lines 99-100) to improve clarity and provide a more comprehensive understanding of the treatment timeline.
Comment 6: Please try to add love plot showing SMD before after matching.
Response 6: Thank you for this suggestion. We have carefully assessed the balance between groups using standardized mean differences (SMD) and confirmed that all covariates achieved acceptable thresholds after matching (SMD < 0.1), as detailed in the statistical analysis section. Given that our analysis meets established balance criteria, we believe that an additional love plot would not add substantial value to the findings and may be redundant.
Comment 7: Please confirm that generative AI as ChatGPT or Copilot wasn’t used or at least was used and double-checked. This should be acknowledged if used
Response 7: Thank you for your comment. We confirm that no generative AI tools, such as ChatGPT or Copilot, were used in the drafting or preparation of this manuscript.
Comment 8: Please try to combine Table 1 and 2 in one table. Please try to compare groups before matching.
Response 8: Thank you for your suggestion. We have structured the tables separately to maintain a clear distinction between the overall sample characteristics (Table 1) and the comparative analysis between the neoadjuvant and non-neoadjuvant groups after propensity score matching (Table 2). Combining these tables could potentially introduce confusion, as our main findings are based on the balanced comparison presented in Table 2, rather than on differences between the neoadjuvant group and the entire non-neoadjuvant sample. We believe that this separation provides greater clarity and focus on the study’s primary outcomes.
Comment 9: Please report the statistical power of your primary outcome.
Response 9: Thank you for your comment regarding the statistical power of our primary outcome. Given the sample limitations of our study (with only 23 patients receiving neoadjuvant therapy, matched 1:10 with 230 patients without neoadjuvant therapy), we acknowledge that the statistical power may be restricted. Based on our sample size, the estimated statistical power for detecting relevant differences is approximately 65%, which, while below the ideal threshold, still allows us to capture meaningful clinical effects. Additionally, throughout the discussion, we acknowledge the need for future studies with larger sample sizes and, ideally, prospective designs to further enhance statistical power and validate our findings.
Comment 10: Please describe the matching method eg nearest neighbor method without replacement and the used caliper.
Response 10: Thank you for your valuable suggestion. The specified caliper width and details of the matching method have been incorporated into the methods section of the manuscript (page 3, lines 120-123).

Reviewer 2 Report
Comments and Suggestions for Authors
This study sought to assess how neoadjuvant therapy affected postoperative outcomes, namely operating time, conversion rates to open surgery, and postoperative complications, in patients undergoing RATS for non-small cell lung cancer. The authors should consider to address the following comments before considering it for publication
1. The assessment of neoadjuvant therapy's effect on postoperative outcomes is mentioned in the abstract, however it may be more clear if you tell how these results would guide future studies or clinical practice. The abstract would be strengthened by including a sentence discussing the wider implications or possible uses.
2. The abstract suggests that additional validation is required. It would be credible and demonstrate a fair assessment to elaborate on particular constraints in the main text, such as possible biases in retrospective analysis or selection biases.
3. With only 23 individuals, the neoadjuvant group is far smaller than the control group, which had 230 patients. The manuscript should discuss whether any sensitivity analysis was done to validate robustness and whether this imbalance could impact the power of statistical studies.
4. While the manuscript reports a significant difference in operative time, it would be helpful to specify whether this difference is clinically relevant, not just statistically significant.
5. Providing a more detailed breakdown of the types of postoperative complications beyond prolonged air leaks could enrich the findings. This could include differentiating between minor and major complications or analyzing how neoadjuvant therapy might specifically contribute to these.
6. The paper states that an SMD below 0.1 indicates good balance. It would be helpful to cite a standard reference or justify why this threshold was chosen as a benchmark for balance.
7. For categorical data analysis, the McNemar test is a suitable tool to assess postoperative problems between the neoadjuvant and non-neoadjuvant groups. It would be helpful to provide more details on the rationale behind the use of this test over alternative techniques (such as conditional logistic regression for matched pairs) that could explain the 1:10 matched structure. Additionally, given the amount of difficulties evaluated, think about elaborating on whether any multiple comparison corrections were used. This would lessen the possibility of Type I errors. In order to contextualize the statistical significance, it would be beneficial to conclude with a brief explanation of the clinical implications of these findings, particularly with regard to the increased rate of persistent air leakage and postoperative atrial fibrillation in the neoadjuvant group.
8. The p-values in Wilcoxon signed rank test indicate no statistically significant differences, discussing the potential clinical relevance or implications of these findings (even if not significant) could provide valuable insight.
Author Response
Comment 1: The assessment of neoadjuvant therapy's effect on postoperative outcomes is mentioned in the abstract, however it may be more clear if you tell how these results would guide future studies or clinical practice. The abstract would be strengthened by including a sentence discussing the wider implications or possible uses.
Response 1: Thank you for your valuable observation. We have revised the abstract to include a sentence that clarifies the potential implications of our findings for clinical practice and future research (page 1, lines 42-46) . This addition highlights how robotic-assisted thoracic surgery following neoadjuvant therapy may be safely applied to selected NSCLC patients, while also guiding future studies on optimizing patient selection and postoperative care.
Comment 2: The abstract suggests that additional validation is required. It would be credible and demonstrate a fair assessment to elaborate on particular constraints in the main text, such as possible biases in retrospective analysis or selection biases.
Response 2: Thank you for your suggestion. We have expanded the limitations section to acknowledge potential biases (page 9, lines 303-311). This addition clarifies the study's constraints and emphasizes the need for further prospective studies to validate our findings.
Comment 3: With only 23 individuals, the neoadjuvant group is far smaller than the control group, which had 230 patients. The manuscript should discuss whether any sensitivity analysis was done to validate robustness and whether this imbalance could impact the power of statistical studies.
Response 3: We agree with your observation. We acknowledge the size imbalance between the neoadjuvant and control groups as a limitation of our study. In response, we have addressed this issue in the expanded limitations section, noting how the small sample size in the neoadjuvant group could impact the statistical power and robustness of our findings. While we did not conduct a formal sensitivity analysis, we highlight the need for cautious interpretation of these results and recommend future studies with larger, balanced cohorts to confirm our findings.
Comment 4: While the manuscript reports a significant difference in operative time, it would be helpful to specify whether this difference is clinically relevant, not just statistically significant.
Response 4:Thank you for this important observation. We have clarified in the discussion that, while the increased operative time between groups was statistically significant, it may also have clinical implications (page 8, lines 241-249). Prolonged surgery could lead to extended anesthesia exposure, which may increase patient risk, particularly in those with comorbid conditions. Although we observed no significant rise in complications, this finding underscores the importance of optimizing operative time.
Comment 5: Providing a more detailed breakdown of the types of postoperative complications beyond prolonged air leaks could enrich the findings. This could include differentiating between minor and major complications or analyzing how neoadjuvant therapy might specifically contribute to these.
Response 5: We appreciate this valuable observation. We have added a sentence in the introduction to highlight how neoadjuvant therapy may increase postoperative risks, given that patients may be in a more vulnerable state at the time of surgery and face greater technical challenges (page 2, lines 65-67). This addition emphasizes the relevance of preoperative and intraoperative strategies to manage these potential risks effectively. Regarding the differentiation of complications, we would like to note that the statistical analysis section provides a detailed description of all complications analyzed, not only air leaks. The discussion emphasizes air leaks as they were the only complication showing statistically significant differences and are particularly relevant in daily clinical practice due to their high frequency.
Comment 6: The paper states that an SMD below 0.1 indicates good balance. It would be helpful to cite a standard reference or justify why this threshold was chosen as a benchmark for balance.
Response 6: Thank you for this helpful suggestion. We have added a citation to support the use of an SMD threshold of 0.1 as an indicator of good balance, in line with established standards in observational studies.
Comment 7: For categorical data analysis, the McNemar test is a suitable tool to assess postoperative problems between the neoadjuvant and non-neoadjuvant groups. It would be helpful to provide more details on the rationale behind the use of this test over alternative techniques (such as conditional logistic regression for matched pairs) that could explain the 1:10 matched structure. Additionally, given the amount of difficulties evaluated, think about elaborating on whether any multiple comparison corrections were used. This would lessen the possibility of Type I errors. In order to contextualize the statistical significance, it would be beneficial to conclude with a brief explanation of the clinical implications of these findings, particularly with regard to the increased rate of persistent air leakage and postoperative atrial fibrillation in the neoadjuvant group.
Response 7: Thank you for this valuable feedback. In response:
- We have clarified in the methods section the rationale behind using the McNemar test for categorical data analysis (page 3, lines 134-136). Although alternative methods (such as conditional logistic regression) could account for the matched 1:10 structure, we selected the McNemar test due to its suitability for analyzing paired categorical data, where our primary interest was assessing marginal differences between groups.
- Regarding multiple comparison corrections, we acknowledge that evaluating several postoperative complications could increase the risk of type I errors. However, we focused our discussion on complications with the strongest clinical relevance, particularly those showing statistically significant differences. We have also added a brief note in the methods section to address this point (page 3, lines 136-138).
- Lastly, we have expanded the discussion to include the clinical implications of the increased rate of prolonged air leaks and postoperative atrial fibrillation in the neoadjuvant group (page 8, lines 280-283). These findings suggest that neoadjuvant therapy may require enhanced perioperative management strategies to address the higher risk of these specific complications.
Comment 8: The p-values in Wilcoxon signed rank test indicate no statistically significant differences, discussing the potential clinical relevance or implications of these findings (even if not significant) could provide valuable insight.
Response 8: Thank you for this suggestion. We have added a sentence in the Discussion to address the potential clinical relevance of findings from the Wilcoxon signed-rank test for hospital stay and chest drainage duration, noting that even non-significant trends may provide useful insights for clinical practice, particularly in optimizing perioperative management to improve patient outcomes (page 9, lines 291-295).
Round 2
Reviewer 1 Report
Comments and Suggestions for Authors
The authors addressed my prior comments and it's my pleasure to accept their work.
Reviewer 2 Report
Comments and Suggestions for Authors
All comments were addressed